# Non-Coding RNA Networks as Potential Novel Biomarker and Therapeutic Target for Sepsis and Sepsis-Related Multi-Organ Failure

**DOI:** 10.3390/diagnostics12061355

**Published:** 2022-05-31

**Authors:** Domenico Di Raimondo, Edoardo Pirera, Giuliana Rizzo, Irene Simonetta, Gaia Musiari, Antonino Tuttolomondo

**Affiliations:** Department of Promoting Health, Maternal-Infant, Excellence and Internal and Specialized Medicine (Promise) “G. D’Alessandro”, University of Palermo, Piazza delle Cliniche, 90100 Palermo, Italy; edoardo.pirera@unipa.it (E.P.); giulianarizzo@yahoo.it (G.R.); irene.simonetta@live.it (I.S.); gaiamusiari@gmail.com (G.M.); bruno.tuttolomondo@unipa.it (A.T.)

**Keywords:** sepsis, biomarker, multi-organ failure (MOF), noncoding RNA, long non-codingRNAs (lncRNAs), circularRNAs (circRNAs), microRNAs (miRNAs)

## Abstract

According to “Sepsis-3” consensus, sepsis is a life-threatening clinical syndrome caused by a dysregulated inflammatory host response to infection. A rapid identification of sepsis is mandatory, as the extent of the organ damage triggered by both the pathogen itself and the host’s immune response could abruptly evolve to multiple organ failure and ultimately lead to the death of the patient. The most commonly used therapeutic strategy is to provide hemodynamic and global support to the patient and to rapidly initiate broad-spectrum empiric antibiotic therapy. To date, there is no gold standard diagnostic test that can ascertain the diagnosis of sepsis. Therefore, once sepsis is suspected, the presence of organ dysfunction can be assessed using the Sepsis-related Organ Failure Assessment (SOFA) score, although the diagnosis continues to depend primarily on clinical judgment. Clinicians can now rely on several serum biomarkers for the diagnosis of sepsis (e.g., procalcitonin), and promising new biomarkers have been evaluated, e.g., presepsin and adrenomedullin, although their clinical relevance in the hospital setting is still under discussion. Non-codingRNA, including long non-codingRNAs (lncRNAs), circularRNAs (circRNAs) and microRNAs (miRNAs), take part in a complex chain of events playing a pivotal role in several important regulatory processes in humans. In this narrative review we summarize and then analyze the function of circRNAs-miRNA-mRNA networks as putative novel biomarkers and therapeutic targets for sepsis, focusing only on data collected in clinical settings in humans.

## 1. Introduction

Sepsis is a life-threatening clinical syndrome caused by a dysregulated host inflammatory response to infection, often associated with multiple organ dysfunction syndrome and death [1]. It is recognized as a leading cause of death worldwide, as highlighted by the Global Burden of Disease, which estimated 48.9 million incident cases in 2017, accounting for 19.7% of all global deaths [2]. The economic and health burden of sepsis worldwide is alarming; mortality in sepsis patients has been estimated to be ≥10%, rising above 40% when evolving to septic shock [1]; in 2011 the total sepsis-related costs for US hospitals accounted for more than US $20 billion [1].

The definition of sepsis has undergone several revisions over the years because of the highly variable clinical spectrum: the 2001 American College of Chest Physicians/Society of Critical Care Medicine (ACCP/SCCM) Consensus Conference Committee criteria for the host systemic inflammatory response syndrome (SIRS) are currently outdated because of their demonstrated poor ability to discriminate between different degrees of clinical severity [3]. Reassessment of these criteria in a clinical setting has shown that they are often found in many inpatients, including those who are noninfectious and who do not proceed to adverse outcomes [3]. The latest definition of sepsis, named “Sepsis-3”, was proposed in 2016 by the SCCM and the European Society of Intensive Care Medicine (ESICM) [1]. According to the SCCM/ESICM, sepsis is defined as a life-threatening organ dysfunction (ascertained as acute change in Sepsis-related Organ Failure Assessment (SOFA score) total score ≥2) “due to a dysregulated host response to infection” [1]. Septic shock is defined as a “subset of sepsis in which particularly profound circulatory, cellular, and metabolic abnormalities substantially increase mortality” [1].

Rapid detection of sepsis is mandatory since the patient’s overall clinical impairment and degree of organ damage are triggered by an extremely complex chain of events involving the recognition of pathogen-associated molecular patterns (PAMPs) of microorganisms by the host immune system [4]. As happens in other acute pathologic conditions such as acute encephalitis [5], but also in several diseases not directly provoked by an infection such as ischemic stroke [6,7,8], atrial fibrillation [9], and many others [10], the characteristics of the interaction between host and pathogen fundamentally affect the degree and severity of the systemic involvement of the patient. The damage-associated molecular patterns (DAMPs) released by the spillover from the injured cells [11] can result in an escalating state of inflammation that can abruptly lead to multiple organ failure (MOF) and can result in death. The prompt initiation of a broad spectrum empirical antibiotic therapy and patient-driven supportive strategies such as fluid resuscitation optimize outcomes. The so-called early goal-directed therapy in the first hour of documented hypotension leads to a 79.9% survival rate, each hour of delay being associated with an average decrease in survival of 7.6% [12]. Despite the development of bedside screening tools to facilitate the early detection of septic patients, a tool to which a definitive diagnostic value can be attributed is still missing, thus the diagnosis is today still challenging, and it continues to depend on the clinical judgment based on nonspecific clinical and laboratory variables. In addition, rapid discrimination between infectious and noninfectious causes presents a daunting challenge. The diagnosis of systemic infection is mainly based on direct microbiological tests such as cultures or polymerase chain reaction-based methods or indirectly using specific immunoglobulin dosage. Unfortunately, microbiology results often take several days to became positive and are not diagnostic in patients with ongoing infection in up to one-third of cases, especially if cultures were collected when antibiotic treatment had already been started [13].

Since the combined sensitivity and specificity of actual biomarkers (e.g., C-reactive protein (CPR), Procalcitonin (PCT) and Interleukin-6 (IL-6)) do not allow for the rapid ascertainment of the diagnosis [14,15] and sepsis-related adverse outcomes rise with every hour of delay of proper intervention, new early biomarkers are urgently needed.

There is a growing amount of data about non-codingRNA, a group of transcripts that do not code proteins at first deemed as redundant RNAs but lately described as highly conserved transcripts involved in gene expression regulation through the modulation of chromatin rearrangement, histone modification, alternative splicing regulation and many other biological processes [16]. Recent findings speculate that circularRNAs (circRNAs), a particular type of long non-codingRNAs (lncRNAs) distinguished by a covalently closed-loop structure with neither 5′ to 3′ polarity nor polyadenosine tail, participates in gene regulation in a different way, regulating the microRNA (miRNAs) concentration in body fluids by competing with several miRNAs and regulating the downstream of messenger RNAs (mRNAs) [17].

Further demonstrating the increasing biological value that non-coding RNAs are proving to have, they seem to play a role in the pathogenesis of different diseases [18], and, given the complex interweaving between circRNAs, miRNAs, lncRNAs and mRNAs, various studies have addressed the issue of their role as novel diagnostic markers and therapeutic targets in many pathologic conditions including sepsis [19,20,21,22,23,24,25]. 

The objective of this review is to summarize the current findings of circRNA-miRNA-lncRNA networks in the context of sepsis as a biomarker and therapeutic target with a focus on their clinical use in a hospital setting and their effectiveness in providing reliable data for the improvement of clinical practice in the adequacy of early diagnosis and treatment of sepsis and septic shock.

## 2. Methodology of Literature Search

A comprehensive literature search was carried out in the MEDLINE database (search terms: “sepsis” + “noncodingRNA”, “miRNA”, “circRNA”, “mRNA”, “lncRNA”, “RNA”, “network”, “biomarker”, “therapy”, “prognosis”, “organ failure”). The search has been restricted to papers published in English without time limit. The authors sought literature by examining reference lists in original articles and reviews. We have included in this review only systematic reviews, metanalyses, randomized trials and randomized controlled trials, selecting studies in which the main objective of the study was the identification and function of circRNA, miRNA, or mRNA networks in the context of novel biomarkers with remarkable prognostic value and/or therapeutic target for sepsis and sepsis-related organ failure.

Each author involved independently evaluated the results of the literature research extracting the most pertinent knowledge, while others verified the accuracy and completeness of the extracted data. Each author made a judgement as to whether the search results were different or confounding, trying to provide as complete an overview of the field as possible to date.

## 3. Role of Biomarkers in Sepsis

According to the Biomarkers Definitions Working Group, a biological marker or biomarker is “a characteristic that is objectively measured and evaluated as an indicator of normal biological processes, pathogenic processes or pharmacologic responses to a therapeutic intervention” [26]. 

A biomarker finds application across four domains or functional classes:As a diagnostic tool, i.e., a biomarker able to confirm a disease;As a tool able to stage or to stratify disease severity;As a prognostic tool;An effective tool for prediction and monitoring of clinical response of an intervention [26].

Biomarkers of sepsis hold the promise of closing the gap in obtaining mycobacterial cultures by providing clinicians with clinically useful data. There is a strong demand for new and accurate sepsis biomarkers, especially in the era of personalized medicine in which physicians must increasingly tailor clinical and therapeutic management to each patient. In the following part of our manuscript, without aiming to address this topic comprehensively, the role of some of the major biomarkers currently used in sepsis will be analyzed by discussing the merits and demerits of their use during the management and treatment of such a serious disease that has its cornerstone in the timeliness of identification and early and appropriate intervention.

### 3.1. C-Reactive Protein (CRP)

C-Reactive Protein (CRP) is a plasma protein belonging to the group of the so-called acute phase reactants which may increase rapidly during inflammatory conditions or secondary to non-specific acute inflammatory stimuli [26]. The acute-phase proteins are produced in the liver during inflammatory states under the control of cytokines: CRP is mainly synthesized through Interleukin-6 (IL-6) and Interleukin-1β (IL-1β) stimulation via the transcription factors STAT3 and NF-κB [27]. CRP, as a component of the innate immune system, during infection may recognize various pathogens associated molecular patterns (PAMPs) such as phospholipid fragments released from damaged cells consequently activating the complement system and finally inducing the death of the targeted cells [28]. CRP is released into the bloodstream after 4–6 h after an inflammatory stimulus and a plasma peak is reached in 36 to 50 h [14,15,16,17,18,19,20,21,22,23,24,25,26,27,28,29]. Several conditions besides infection can result in the elevation of the CRP-serum level [30]; a meta-analysis of Simon et al. [31] demonstrated a sensitivity of 75% [95% CI: 62–84%] and a specificity of 67% [95% CI: 56–77%] for CRP in differentiating bacterial infection from the noninfective cause of inflammation.

Liu et al. [32], in a systematic review and meta-analysis including 45 studies and 5654 patients, showed an acceptable level of sensitivity of 75% (95% CI: 69–79%) but a weak level of specificity of 67% (95% CI: 58–74%) for the ability of CRP to differentiate patients with sepsis vs. non-infectious inflammatory state/disorders. Tan et al. [33], comparing the ability of CRP and PCT to serve as biomarkers for sepsis diagnosis show similar sensitivity (CRP: 80%, 95% CI: 63–90%, procalcitonin: 80%, 95% CI: 69–87%) but significantly lower specificity for CRP (61%; 95% CI: 50–72%) than procalcitonin (77%; 95% CI: 60–88%) [33]. A possible explanation for the lower diagnostic accuracy of CRP as a sepsis biomarker (low specificity and moderate sensibility) could account for the slow-release kinetics as a consequence of the inflammatory stimulus and its increase also due to other pathological conditions besides infections (e.g., trauma, burns, surgery or various immune-inflammatory conditions [34,35].

Finally, the limits showed that CRP remains a widely used diagnostic and therapeutic biomarker in sepsis to date, mainly because a decrease in its values correlates with the success of antimicrobial treatment [36].

### 3.2. Procalcitonin (PCT)

Procalcitonin is the precursor of calcitonin, released by the C-cells of parathyroid glands. Assicot et al. [37] in 1993 for the first time described the association between PCT serum levels and severe bacterial infection. Compared to CRP, the PCT has a better kinetic profile, increasing within 3–6 h after the onset of infection reaching its serum peak after 6–8 h [29]. Several studies investigated the diagnostic performance of PCT. A meta-analysis of Uzzan et al. [35], including studies from 1996 to 2004, showed a higher accuracy of PCT levels than CRP levels for the diagnosis of sepsis (Global diagnostic accuracy odds ratios: CRP 5.43 [95% CI: 3.19–9.23] vs. PCT 14.69 [95% CI 7.12–30.27] [35]. However, the authors included a restricted cohort study based only on surgery or trauma patients, and thus the conclusion cannot be extended to patients other than surgical conditions [35].

Tang et al. [38], in a meta-analysis of 18 studies, pointed out that PCT was not adequate in discriminating between sepsis and SIRS (both sensitivity and specificity were 71% [95% CI: 67–76] and the Area Under the Summary Receiver Operator Characteristic Curve was 0.78 [95% CI: 0.73–0.83] [38]. Another meta-analysis of 30 observational studies evaluating 3244 mixed subjects (pediatric and adult patients admitted in the Intensive Care Unit or Emergency Room), has given the PCT a sensitivity of 77% [95% CI: 72–81%] and a specificity of 79% [95% CI: 74–84%], with AUC 0.85 [95% CI 0.81–0.88] for accuracy in discriminating sepsis from a non-infectious state [39].

Several studies have also confirmed the clinical utility of PCT in driving antimicrobial therapy surveillance and the eventual de-escalation of antibiotic treatment [14,15,16,17,18,19,20,21,22,23,24,25,26,27,28,29].

To date, there are no established cut-off values of serum PCT concentrations that are able to discriminate sepsis versus septic shock [29].

### 3.3. Presepsin

Presepsin, the N-terminal fragment of 13 kDa of the sCD14 (the soluble form of the receptor of lipopolysaccharide-lipopolysaccharide binding protein), is an emerging biomarker and early indicator of bacterial infections [40]. Presepsin, as part of the Toll-like receptor group, takes part of the innate immune system, binding several pathogen-associated molecular patterns (PAMPs) such as lipopolysaccharide (LPS) of Gram- or peptidogligans [40]. In recent years, sCD14 has become one of the most widely sepsis biomarkers studied: the level of sCD14 increased significantly in patients with sepsis and septic shock compared with healthy people, and the change was significantly related to the severity and prognosis of the disease [29,41,42,43,44]. The diagnostic power of presepsin in detecting sepsis showed with a pooled sensitivity of 77–86% and a specificity of 73–78% [41,42,43,44]. Nevertheless, presepsin still needs wider investigation and further validation and comparison with standard sepsis biomarkers prior to being recommended for the hospital-setting.

## 4. Non-CodingRNA

Non-codingRNA, including long non-codingRNAs (lncRNAs), circularRNAs (circRNAs) and microRNAs (miRNAs), take part in a complex chain of events playing a pivotal role in several important regulatory processes in humans. Non-codingRNAs are traditionally classified based on the length of nucleotides (nt) in small non-codingRNAs (sncRNAs) and long non-codingRNAs. The ncRNAs with 200 nt or lower are referred to as sncRNAs including microRNAs (miRNAs), small interfering RNAs (siRNAs) and piwi-interactingRNAs (piRNAs) whereas those with 200 nt or higher are referred as lncRNAs, such as promoter-associated transcripts (PATs), enhancerRNAs (eRNAs) and circularRNAs (circRNAs).

Numerous studies have demonstrated that exosomes, phospholipid bilayer vesicles that originate from the membrane vesicles of the endosomes and are secreted by almost all cells, contain a variety of functional molecules that are crucial mediators of intercellular communication including lncRNAs, circRNAs and miRNAs. After exosomes are released into the tissue fluid, they arrive at the target cells and begin to deliver the different molecules contained (such as circRNAs), thus initiating functional responses and inducing subsequent phenotypic changes [45,46,47]. Recent studies have shown altered expressions of several non-coding RNAs such as lncRNAs, circRNAs and miRNAs during sepsis. Interestingly, noncoding RNAs have also been found to participate in the pathogenesis of multiple organ system failure through different mechanisms. In the following section of the review, the role of these three classes of noncoding RNAs in the pathophysiology of sepsis and sepsis-related multi-organ failure (MOF) will be examined.

### 4.1. CircRNAs

CircRNAs are an endogenous non-codingRNA which have as their main characteristic a closed loop structure with a covalent bond linking the 3′ to 5′ ends [48,49]. 

CircRNAs are highly conserved molecules expressed in a broad range of human cells, both in physiologic and pathologic conditions [50]. Memczak et al. detected 1950 circRNAs in HEK293 cells, 1903 circRNAs in the brain and fetal cells of mouse and through 724 circRNAs from different developing stages of Caenorhabditis Elegans (a nematode worm about 1 mm in length) [48]. 

Enuka et al. [51] estimate that circRNAs have a half-life of 18.8–23.7 h, roughly 2.5-fold higher that their linear form [52]. Also, circRNAs are insensitive to the common degradation pathways (i.e., RNAase or RNAexonucleases) [53]. Mature circRNAs are usually found in cytoplasms, whereas immature circRNAs that are still susceptible to intronic splicing remain in the nucleus [54].

#### 4.1.1. CircRNAs Biogenesis

CircRNAs are primarily generated by the transcription of exonic and/or intronic sequences of a protein-coding gene in a complex reaction catalyzed by the RNA polymerase II (RNApol II). Both circRNAs and mRNAs, in their linear forms, are produced by the same precursor or pre-mRNAs undergoing a totally different splicing mechanism [55]. CircRNAs are produced by spliceosome machinery through a back-splicing process, resulting in a covalently closed loop structure between the 3′ upstream splice site and the 5′ downstream splice site [56,57,58,59,60,61]. 

#### 4.1.2. CircRNAs Functions

CircRNAs act by separating miRNAs from their target mRNAs, thereby influencing miRNA-mediated gene suppression or expression. The interaction between circRNA-miRNA appears to be critical to the optimal functionality of our organism [62,63]. CircRNAs may act as a dynamic scaffold influencing protein interactions having the potential to regulate protein function by binding, storing, sequencing and isolating proteins to specific subcellular locations [64]. Finally, nuclear circRNAs may act as regulators of the transcription by promoting the extended activity of RNA polymerase II [46,65] or through other alternative pathways which are still being studied.

Considering all of these features, the tissue development specific expression patterns and the putative crucial regulatory functions in various diseases, circRNAs thus have all of the qualities to be used as novel biomarkers in several pathologic conditions [19,21,66,67].

### 4.2. Long Non-CodingRNAs (lncRNAs)

The lncRNAs (200 nt or more) belong to the large class of non-codingRNAs that perform housekeeping functions in numerous biological processes through the regulation of gene expression at the post-transcriptional and transcriptional level [68,69,70]. Changes in the expression levels of lncRNAs affect the malignancy phenotype in various types of cancer such as colorectal, lung, liver, breast, ovarian cancers and leukemia [71,72], and are linked to development state [73] and may be observed during various phases of T-cell differentiation [74]. 

#### 4.2.1. lncRNAs Biogenesis

To date, the lncRNAs biogenesis is not completely clarified. Transcriptome-wide studies show that lncRNAs biogenesis owns a peculiar expression pattern that is cell type-specific and stage-specific [75,76], which is also regulated by cell type and stage-specific stimuli [77] Briefly, lncRNAs are widely interspersed in the genome. Enhancers, promoters, and intragenic regions are the main DNA elements from which lncRNAs are transcribed [78].

#### 4.2.2. lncRNAs Functions

The molecular function of lncRNAs may be summarized as four archetypes: (1) signal: lncRNAs can serve as molecular signal able to activate/silence specific genome sequences by interacting with chromatin and recruiting the chromatin modifying systems [76]; (2) Decoy: lncRNAs play a central role in the regulation of genome transcription, mainly by inhibiting it but occasionally also by activating it [76,77]; (3) Guide: lncRNAs can modify the chromatin structure guiding specific proteins to specific targets that ultimately cause gene silencing [76,77]; (4) Scaffold: lncRNAs can serve as a scaffold for assembling two or more proteins by inducing changes in chromatin as a consequence. They may therefore play a role in the activation or transcriptional silencing of specific genome sequences [76,77].

### 4.3. MicroRNAs (miRNAs)

miRNAs are endogenous non-coding transcripts of 19–22 nucleotides that modulate the translation of target mRNAs at a post-transcriptional level [79].

#### 4.3.1. miRNAs Biogenesis

The RNA polymerase II transcribes a primary miRNA called pri-miRNA (~500–3000 nt); starting with this precursor a premature miRNA (pre-miRNA) is formed [80,81]. Through exportin 5, the pre-miRNA is exported in the cytoplasm and an miRNA duplex is processed. The “miRNA duplex” is bound into the RISC (RNA-induced silencing complex) with the final release of the mature miRNA [80]. As previously mentioned, circRNAs sequester mature miRNAs, acting as an miRNA sponge, through the interaction between the RNA binding proteins, diminishing miRNA functions.

#### 4.3.2. miRNAs Functions

miRNAs are believed to be among the most important regulators of cellular communication, playing a pivotal role in maneuvering the linear RNA’s expression at various levels. The most notably and predominant function of miRNAs is the binding of complementary sequences of the target linear RNAs that results in the inhibition of mRNA translation [82].

## 5. Non-Coding RNA and Sepsis

During sepsis the immune system is widely activated; the level and extent of the immune response triggered by the pathogen is different from subject to subject and depends largely on his/her state of immunocompetence [83,84].

Studies performed mostly in experimental animal models investigating the role of non-codingRNAs in the modulation of inflammation have identified some interesting networks that have been shown to be actively involved. Accumulating evidence shows that lncRNAs and miRNAs are involved in the sepsis inflammatory response, but the role played by different non-coding RNA networks in different biological contexts is extremely complex, variable, and involves the intervention of several mediators and effectors. An example will make clear the complexity of the field of research we are discussing: lncRNA taurine upregulated gene 1 (lncRNA TUG1) seems to participate in several pathophysiological processes. The overexpression of TUG1 has demonstrated the ability to alleviate the inflammatory response (including apoptosis and autophagy) in acute lung injury [85]. These relevant clinical effects seem to be mediated via targeting miR-34b-5p [86] and miR-27a-3p [87]. On the other hand, TUG1 silencing reduces the inflammation and apoptosis of renal tubular cells in an ischemia-reperfusion model via targeting of miR-449B-5p [88], and regulates the expression of various genes such as matrix metalloproteinase [89], protects against myocardial ischemia upregulating miR 142-3p [90] and downregulating miR-29a-3p [91] through the modulation of the miR-532-5p/Sox8 axis [92] and the miR-145-5p-Binp3 axis [93,94]. These are just a few of the many and complex interactions highlighted to date. 

Despite the mentioned challenge of accurately defining the roles played by specific non-coding RNAs, some research has helped us identify some molecules of likely clinical interest. LncRNA metastasis-associated lung adenocarcinoma transcript 1 (MALAT1) has been linked to sepsis. MALAT1 downregulation inhibits the LPS-induced inflammatory response by preventing the release of IL-6 and TNF-α and the NF-κB signaling pathway by upregulating miR-150-5p [95]. The MALAT1/miR-214/Toll-like receptor (TLR)5 signaling pathway dysregulation enhances the risk of post-burn sepsis by promoting greater production of proinflammatory cytokines [96]. LncRNA nuclear-enriched abundant transcript 1 (NEAT1) seems to also be involved in sepsis progression. In rats, NEAT1 knockdown could significantly improve the sepsis-induced myocardial injury preventing cardiac insufficiency and consensually increasing the ejection fraction (*p* < 0.05) [97]. Furthermore, another study indicates that NEAT1 inhibits the LPS-induced progression of sepsis in RAW264.7 cells by modulating the miR-31-5p/POU2F1 axis [98], suggesting that NEAT1, which also positively correlates with Th1 and Th17 levels [99], will be a potential target for clinical treatment of sepsis-induced organ damage.

Several miRNAs have been found to influence the course of sepsis, providing another substantial amount of evidence [100]. Over a hundred of different miRNAs have been investigated [101], identifying more or less prominent roles in the pathophysiology of sepsis-related injury by influencing the level of inflammasome induced.

What emerges is therefore an extremely interconnected network in which it is difficult to disentangle, especially if the objective is, as in our case, to try to determine whether there are the premises to attribute the role of biomarker of sepsis to one or more of the non-coding RNAs analyzed so far. It is a picture in which for each different pathological condition studied, different networks of non-coding RNA play different roles, making it impossible to assign a unique label to a specific lncRNA or miRNA.

## 6. Assessment of the Clinical and Prognostic Value of Non-Coding RNAs as Biomarkers in Sepsis

The role actually played in sepsis by the many non-coding RNAs that have been studied so far is only partially clarified. Many studies using cell and animal models have tried to demonstrate that non-coding RNAs may be used to control the multi-organ damage due to the septic process, but considerable challenges must be overcome in order to successfully translate these approaches into clinical practice. The concrete risk is to discuss notions that remain theoretical, not having the clinical relevance to propose themselves as concrete novelties in the future day-to-day clinical practice. Attempting to provide indications that are as translatable as possible into clinical practice, the following will be listed only the evidence available to date in which it was assessed the usefulness of miRNAs in sepsis in human clinical settings.

### 6.1. circRNAs

Using the research criteria previously described, only three studies to date have evaluated circ-RNAs by addressing the issue of their reliability as biomarker and prognostic value in clinical settings of sepsis in humans.

Wei et al. [102] explored the clinical values of circular RNA protein kinase C iota (circ-PRKCI) and its target microRNA-545 (miR-545) in sepsis patients. Plasma samples of 121 sepsis patients and 60 healthy controls were collected. Decreased circ-PRKCI expression and increased miR-545 expression were observed in sepsis patients compared to healthy controls, both of which had close correlations with sepsis risk. Decreased circ-PRKCI and increased miR-545 expressions were associated to 28-day mortality risk in sepsis patients, which were slightly lower than the predictive values of APACHE II score and SOFA score for predicting 28-day mortality risk. 

Tian et al. [103] studied changes in circRNA expression in exosomes by circRNA microarray analysis in sepsis patients. ROC analysis showed that hsa_circRNA_104484 and hsa_circRNA_104670 have the potential to be used as novel diagnostic biomarkers and molecular therapeutic targets for sepsis. 

Hong et al. [104] evaluated circFADS2 expression, a circRNA with putative protective roles in LPS-induced inflammation. Their results suggest that CircFADS2 is upregulated in sepsis to suppress LPS-induced lung cell apoptosis. 

### 6.2. lncRNAs

Using the research criteria previously described, twelve studies addressed the issue of reliability as a biomarker and prognostic tool of lncRNAs in clinical settings of sepsis in humans. Table 1 shows the main findings as well as the diagnostic power of the different lncRNAs evaluated.

### 6.3. miRNAs

The research that has evaluated miRNAs with this aim in clinical settings of sepsis are of significant number, attracting the interest of researchers. We have tried to summarize in Table 2 the main elements of all available studies and in Figure 1 the main regulatory mechanisms of the pro-inflammatory state and interaction with host defense mechanisms that may substantiate their role as biomarkers in sepsis.

## 7. Conclusions

When facing sepsis or septic shock, time management is crucial to ensure the maximum chance of survival for the patient. Since the combined sensitivity and specificity of actual biomarkers (e.g., CPR, PCT, IL-6, etc.) do not allow for the rapid ascertainment of the diagnosis [14,15], they cannot always discriminate early enough between infectious and non-infectious patients and rapidly evolving patients from the more stable ones, whilst sepsis-related adverse outcomes rise with every hour of delay of proper intervention; new early biomarkers are urgently needed. 

Non-codingRNA, including lncRNAs, circRNAs and miRNAs, cooperate in a comprehensive network deeply involved in gene function modulation, playing a pivotal role in several important regulatory mechanisms in humans, including the pathophysiology of sepsis and the sepsis-associated organ dysfunction. 

The numerous studies carried out in the field of non-coding RNAs, their biogenesis, and their biological and clinical significance in recent years have identified the finding of the selective enrichment of exosomes with various networks of non-codingRNAs, different in health and disease according to options and mechanisms almost unknown today. This issue is certainly one of those most deserving of attention and investigation in the future.

Among the many studies conducted in clinical settings to ascertain the effective role of specific non-coding RNAs in supporting the early diagnosis of sepsis and in guiding therapeutic management, some have provided data that are worthy of highlighting and may provide a concrete starting point for future investigations. Some lncRNa, such as NEAT1, MALAT1, and HULC have shown interesting and promising potentials, as well as many dozens of miRNAs that seem to have the potential to flank, if not replace, the role of biomarkers in the molecules that we currently use today.

Much remains to be investigated and written about non-coding RNAs as a prognostic biomarker for sepsis. Targeted clinical studies aimed at identifying their role in everyday practice more accurately are therefore needed in the coming years. 

## Figures and Tables

**Figure 1 diagnostics-12-01355-f001:**
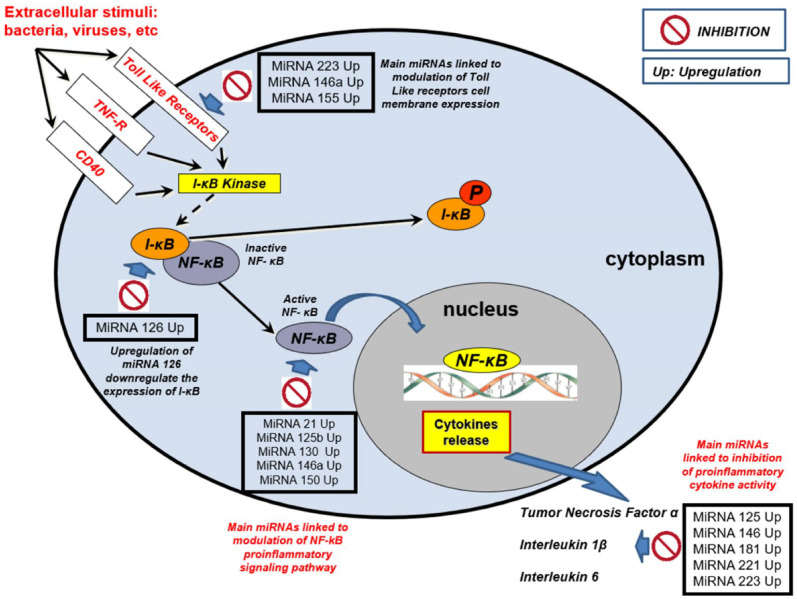
MicroRNAs (miRNAs) in sepsis. Regulatory mechanisms of the pro-inflammatory state and interaction with host defense mechanisms that may substantiate their role as biomarkers.

**Table 1 diagnostics-12-01355-t001:** Main findings of the study regarding lncRNAs addressing the issue of their reliability as biomarker and prognostic tool in clinical settings of sepsis in humans.

Ref.	Year	lncRNA	Novel/Validation	Pattern of Expression	Sample Size			Diagnostic Power			Other Results
					Sepsis Cases	Control	N°	1. Sensitivity (%)	2. Specificity (%)	AUC	
[105]	2019	lncRNA ITSN1-2	Validation	Upregulated	309	HC	300	59.5	86.3	0.777	Positive correlation with APACHE II, CRP, TNF-α, IL-6, il-8; Negative correlation with IL-10
[106]	2019	lncRNA ZFAS1	Novel	Downregulated	202	HC	200	NR	NR	0.814	Negative correlation with APACHE II, CRP, TNF-α, IL-6; Positive correlation with IL-10; predicts survivor from non-survivor
[107]	2019	lncRNA ANRIL	Novel	Upregulated	126	HC	126	NR	NR	0.800	Positive correlation with CRP, PCT, APACHE II, SOFA, TNF-α, IL-8
[108]	2019	lncRNA MALAT1	Novel	Upregulated	190	HC	190	NR	NR	0.823	Positive correlation with PCT, Scr, WBC, CRP, SOFA and APACHE II; predict 28-day mortality
[109]	2020	lncRNA THRIL	Novel	Upregulated	32 ARDS +sepsis	nonARDS-sepsis	77	NR	NR	0.706	Positive correlation with CRP, PCT, TNF-α, IL-1β
[110]	2020	lncRNA GAS5	Novel	Downregulated	60	HC	60	NR	NR	NR	Positive correlation with miRNA-214
[111]	2020	lncRNA MEG3	Validation	Upregulated	112	HC	100	77.7	94	0.893	Predictive role for ARDS-sepsis
[112]	2020	lncRNA MALAT1	Validation	Upregulated	120	HC	60	NR	NR	0.910	Positive correlation with PCT, Lactate levels, SOFA and APACHE II
[113]	2020	lncRNA MALAT1	Validation	Upregulated	196	HC	196	91.3	78.6	0.931	Negative correlation with miR125a and albumin; positive correlation with APACHE II, SOFA, Scr, CRP, IL-6, IL-8, IL-1β, TNF-α
[114]	2020	lncRNA NEAT1	Validation	Upregulated	102	HC	100	NR	NR	0.992	Negative correlation with miR-125a
[115]	2021	lncRNA HULC	Novel	Upregulated	174	HC	100	78.7	97	0.939	Positive correlation with TNF-α, IL-6, IL-17, ICAM1, and VCAM1 APACHE II, SOFA Score,
[116]	2021	lncRNA PVT1	Validation	Upregulated	109	HC	100	NR	NR	NR	Predictive role for ARDS and 28-day mortality, positive correlation with disease severity;

Novel: lncRNA identified for the first time; Validation: confirmation of a finding already reported in the literature. Ref: reference; NR: Not Report; HC: Healthy controls; IL: Interleukin; TNF-α: Tumor necrosis factor-α; SOFA: Sequential Organ Failure Assessment; APACHE II: Acute Physiologic Assessment and Chronic Health Evaluation II; PCT: procalcitonin; CRP: C-reactive protein; Scr: serum creatinine; WBC: white blood count; AUC: Area under the curve; miR: microRNA; ARDS: Acute respiratory Distress Syndrome.

**Table 2 diagnostics-12-01355-t002:** Main findings of the study regarding miRNA addressing the issue of their reliability as biomarkers and prognostic tools in clinical settings of sepsis in humans.

Ref.	Year	miRNA	Novel/Validation	Pattern of Expression	Sample Size			Diagnostic Power			Other Results
					Sepsis Cases	Control	Number	Sensitivity (%)	Specificity (%)	AUC	
[117]	2009	miR-146a	Novel	Downregulated	50	SIRS + HCs	30 + 20	NR	NR	0.804	N/A
[117]	2009	miR-223	Novel	Downregulated	50	SIRS + HCs	30 + 20	NR	NR	0.858	N/A
[118]	2012	miR-15a	Novel	Downregulated	166	SIRS	32	68.3	94.4	0.858	N/A
[119]	2013	miR-150	Novel	Upregulated	23	SIRS	22	72.7	85.7	0.830	N/A
[119]	2013	miR-4772-5p-iso	Novel	Downregulated	23	SIRS	22	68.2	71.4	0.760	N/A
[120]	2013	miR-146a	Validation	Downregulated	14	SIRS	14	60	87.5	0.813	N/A
[121]	2013	miR-146a	Validation	Upregulated	40	SIRS	20	77.5	77	0.815	Positive correlation with miR-223, IL-10, TNF-α
[121]	2013	miR-123	Novel	Upregulated	40	SIRS	20	77.5	55	0.678	Positive correlation with miR-146a, IL-10, TNF-α
[122]	2014	miR-25	Novel	Downregulated	70	SIRS	30	NR	NR	0.806	Negative correlation with SOFA, PCT, CRP. Predictive role in 28-day mortality risk (AUC: 0.756)
[123]	2014	miR-155	Novel	Upregulated	60	HCs	30	NR	NR	NR	Positive correlation with SOFA; predictive role in 28-days mortality risk (AUC: 763)
[124]	2014	miR-143	Novel	Upregulated	103	SIRS	95	78.6	91.6	0.910	Positive correlation with SOFA, APACHE II
[125]	2015	miR-499	Novel	Upregulated	112	HCs	20	86.7	90.8	0.838	N/A
[126]	2016	miR-223	Validation	Upregulated	187	HCs	186	56.6	86.6	0.754	Positive correlation with CRP, TNF-α, IL-1β, IL-6, IL-8 and negatively with IL-10; predicts survivor from non-survivor
[127]	2016	miR-155-5p	Validation	Upregulated	105	HCs	35	85.3	80.6	0.855	N/A
[127]	2016	miR-133a-3p	Novel	Upregulated	105	HCs	35	97.9	54.8	0.769	N/A
[128]	2017	miR-328	Novel	Upregulated	110	HCs	89	87.6	86.4	0.926	Positive correlation with Scr, WBC, CRP, PTC, APACHE II, SOFA,
[129]	2017	miR-495	Novel	Downregulated	105	HCs	100	89.5	83	0.915	Distinguishes sepsis from sepsis shock (Sen: 85.3%; Spec: 87.3; AUC 0.885); Negative correlation with Scr, WBC, CRP, PCT, APACHE II, SOFA
[130]	2017	miR-7110-5p	Novel	Upregulated	44	Non sepsis pneumonia + HC	96	84.2	90.5	0.883	N/A
[130]	2017	miR-223-3p	Validation	Upregulated	44	Non sepsis pneumonia + HCs	96	82.9	100	0.964	N/A
[131]	2017	miR-19b-3p	Novel	Downregulated	103	HCs	98	85.4	85.7	0.921	Independent prognostic factor for 28-days survival; Negative correlation with IL-6, TNF-α
[132]	2018	miR-126	Novel	Upregulated	208	HCs	210	NR	NR	0.726	Positive correlation with APACHE II, ICU stay, MCD, Scr, CRP, TNF-α, IL-6, IL-8 and negative with IL-10
[133]	2018	miR-122	Validation	Upregulated	108	Non sepsis infection	20	58.3	95	0.760	Independent prognostic factor for 30-days mortality (HR: 4.3)
[134]	2018	miR-10a	Novel	Downregulated	62	HCs	20	NR	NR	0.804	Negative correlation with APACHE II, SOFA, CRP, PCT; predictive role in 28-days mortality risk (AUC: 0.795)
[135]	2018	miR-125b	Novel	Upregulated	120	HCs	120	49.2	80	0.658	Positive correlation with APACHE II, SOFA, Scr, CRP, PCT, TNF-α, IL-6; Independent factor for mortality risk. In this study miR-125a upregulation was not associated with enhanced disease severity, inflammation, and increased mortality in sepsis patients
[136]	2018	miR-146a	Validation	Downregulated	55	HCs	60	86.6	56.6	0.803	Negative correlation with CRP, PCT, IL-6, TNF-α
[137]	2018	miR-181a	Novel	Downregulated	102	Local infection	50	83.3	84	0.893	N/A
[138]	2018	miR-101	Novel	Upregulated	50	SIRS	30	84	84	0.908	N/A
[138]	2018	miR-187	Novel	Upregulated	50	SIRS	30	72	76	0.789	N/A
[138]	2018	miR-21	Novel	Upregulated	50	SIRS	30	64	66	0.711	N/A
[139]	2019	miR-494-3p	Novel	Downregulated	NR	HCs	NR	NR	NR	0.837	N/A
[140]	2019	miR-122	Novel	Upregulated	25	LWI	25	100	100	1.000	Higher AUC than CRP and WBC; 56% of accuracy as a prognostic biomarker
[141]	2019	miR-21	Validation	Downregulated	219	HCs	219	NR	NR	0.801	Negative correlation with APACHE II, SOFA, Scr, CRP, TNF-α, IL-1β, IL-6, IL-17;
[142]	2019	miR-103	Novel	Downregulated	196	HCs	196	NR	NR	NR	Negative correlation with APACHE II, SOFA, Scr, CRP, TNF, IL-1β, IL-6, IL-8 positive with albumin; predicted high ARDS risk (AUC: 0.727) and increased 28-days mortality risk (AUC: 0.704)
[142]	2019	miR-107	Novel	Downregulated	196	HC	196	NR	NR	NR	Negative correlation with APACHE II, SOFA, Scr, CRP, TNF, IL-1β, IL-6, IL-8 positive with albumin; predicted high ARDS risk (AUC: 0.694) and increased 28-days mortality risk (AUC: 0.649)
[143]	2019	miR-146a	Validation	Upregulated	180	HCs	180	NR	NR	0.774	Positive correlation with APACHE II, SOFA, Scr, CRP, TNF-α, IL-1β, IL-6, IL17 and negative with albumin
[143]	2019	miR-146b	Novel	Upregulated	180	HCs	180	NR	NR	0.897	Good predictive value in 28-days mortality risk (AUC: 0.703);Positive correlation with APACHE II, SOFA, Scr, CRP, TNF-α, IL-1β, IL-6, IL17 and negative with albumin
[144]	2019	miR-125a	Novel	Upregulated	150	HCs	150	NR	NR	0.749	Positive correlation with APACHE II, SOFA. Not correlates with level of inflammation, disease severity, and 28-day mortality risk in sepsis patients
[144]	2019	miR-125b	Validation	Upregulated	150	HCs	150	NR	NR	0.839	Positive correlation with APACHE II, SOFA, CRP, TNF-α, IL-6, IL-17, IL-23; predictive role in 28-days mortality risk (AUC: 0.699)
[145]	2019	miR-210	Novel	Upregulated	125	HCs	110	81	80.9	0.852	Positive correlation with BUN, Scr, CysC
[145]	2019	miR-494	Validation	Upregulated	125	HCs	110	80.9	72.1	0.847	Positive correlation with BUN, Scr, CysC
[145]	2019	miR-205	Novel	Upregulated	125	HCs	110	78.6	90.5	0.860	Negative correlation with BUN, Scr, CysC
[146]	2020	miR-452	Novel	Upregulated	97	HCs	89	NR	NR	NR	High efficacy in distinguishing AKI in sepsis patients
[147]	2020	miR-125a	Validation	Upregulated	41	noARDS-sepsis	109	NR	NR	0.650	Positive correlation with Scr, APACHE II, SOFA
[147]	2020	miR-125b	Validation	Upregulated	41	noARDS-sepsis	109	NR	NR	0.739	Positive correlation with with Scr, APACHE II, SOFA
[148]	2021	miR-29c-3p	Novel	Upregulated	86	HCs	85	80.2	81.1	0.872	Positive correlation with APACHE II, SOFA, CRP, PCT

NOVEL (identified for the first time); VALIDATION (confirmation of a finding already reported in the literature). Ref: reference; NR: data not reported; N/A: not applicable; HC: Healthy controls; LWI: local wound infection; MCD: mechanical ventilation duration; SIRS: systemic inflammatory response syndrome; IL: Interleukin; TNF-α: Tumor necrosis factor-α; SOFA: Sequential Organ Failure Assessment; APACHE II: Acute Physiologic Assessment and Chronic Health Evaluation II; PCT: procalcitonin; CRP: C-reactive protein; Scr: serum creatinine; WBC: white blood count; BUN: blood urea nitrogen; CysC: Cystatin C; AUC: Area under the curve; Sen; sensibility; Spec: Specificity; miR: microRNA; ICU: Intensive Care Unit; HR: Hazard Ratio; ARDS: Acute respiratory Distress Syndrome; AKI: Akute Kidney Injury.

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
