# Peer review of "Non-Coding RNA Networks as Potential Novel Biomarker and Therapeutic Target for Sepsis and Sepsis-Related Multi-Organ Failure"

_diagnostics, 2022, doi:10.3390/diagnostics12061355_

Round 1

Reviewer 1 Report

In the present work, the authors make an extensive review of the possible markers for the early diagnosis of sepsis and sepsis related multi-organ failure. The topic is extremely important and, as the authors state, there is still a long way to go to achieve a marker with optimal sensitivity and specificity.

However, in its current format, the manuscript is very difficult to read and would need to be improved to be of any use to readers.

First of all, English should be reviewed by a native English speaker. There seems to be many sentences with some grammatical errors, verbs lacking, articles missing, etc...).

The amount of data offered in the text is very difficult to assimilate. An effort should be made to summarize the most relevant ones in easier-to-read tables and not replicate them in the text. The text itself is excessively long and should be rationally shortened.

Many typos should be corrected throughout the text.

As a few examples:

Introduction.

Line 36. In the sentence “representing 19.7 of all global death”, please write “19.7%”.

Line 41. “Undergone to several changes”. Please, consider erasing “to”.

Line 43. “Outdated” instead of “lapsed” (?).

Lines 48-54. The sentence should be grammatically checked.

Line 74. Given that it is the first time it appears, please explain PCR- (Polymerase chain reaction?).

Line 78. “Sensibility and sensitivity”. Do you mean “sensitivity and specificity”?

Line 97. Simplify to “[19-25]”.

Line 100. “performance”, instead of “per-formance”.

  1. Role of Biomarkers in Sepsis

Line 139. “aspecific”. Do you mean “unspecific”?

Lines 140 and 144. Be consistent with the placement of the bracket, preferably before the punctuation sign.

Line 147. “targeted cells [28] Its pharmacokinetics” Punctuation sign?

Line 150. “Simon et Al showed…”. Consider, please, “Simon et al. showed…”

Line 154. 75% (95% CI 0.69-0.79),.. Please, be consistent: % or ratio (0.75).

 … Etc. (impossible to check all of them, sorry).

Author Response

REVIEWER #1

We would like to thank you for the careful and thorough review of our manuscript. We greatly appreciate the effort you made concerning your critique for the review of our study. We have accepted all your suggestions and revised the article according to them.

In the present work, the authors make an extensive review of the possible markers for the early diagnosis of sepsis and sepsis related multi-organ failure. The topic is extremely important and, as the authors state, there is still a long way to go to achieve a marker with optimal sensitivity and specificity.

However, in its current format, the manuscript is very difficult to read and would need to be improved to be of any use to readers.

First of all, English should be reviewed by a native English speaker. There seems to be many sentences with some grammatical errors, verbs lacking, articles missing, etc...).

We deeply regret the unjustifiable number of typographical errors that were present in the original version of the manuscript. We have provided as suggested for a full revision of the manuscript and a re-editing of the language by a native English speaker. We trust that the revised version can be judged favorably.

The amount of data offered in the text is very difficult to assimilate. An effort should be made to summarize the most relevant ones in easier-to-read tables and not replicate them in the text. The text itself is excessively long and should be rationally shortened.

Thank you for the suggestion. The text has been lightened in all sections, a new table has been created (with the aim of replacing some of the information cited in the text, which has then been consensually removed); all sections of the text have been rationalized for ease of reading.

Many typos should be corrected throughout the text.

As a few examples:

Introduction.

Line 36. In the sentence “representing 19.7 of all global death”, please write “19.7%”.

Line 41. “Undergone to several changes”. Please, consider erasing “to”.

Line 43. “Outdated” instead of “lapsed” (?).

Lines 48-54. The sentence should be grammatically checked.

Line 74. Given that it is the first time it appears, please explain PCR- (Polymerase chain reaction?).

Line 78. “Sensibility and sensitivity”. Do you mean “sensitivity and specificity”?

Line 97. Simplify to “[19-25]”.

Line 100. “performance”, instead of “per-formance”.

  1. Role of Biomarkers in Sepsis

Line 139. “aspecific”. Do you mean “unspecific”?

Lines 140 and 144. Be consistent with the placement of the bracket, preferably before the punctuation sign.

Line 147. “targeted cells [28] Its pharmacokinetics” Punctuation sign?

Line 150. “Simon et Al showed…”. Consider, please, “Simon et al. showed…”

Line 154. 75% (95% CI 0.69-0.79),.. Please, be consistent: % or ratio (0.75).

 … Etc. (impossible to check all of them, sorry).

All the typos indicated (and also the many not indicated) have been corrected. We apologize again to the reviewer for the understandable reading difficulties that resulted from the poor quality of the original manuscript.

We hope that we have successfully changed our manuscript according to your suggestions and that we have provided all the necessary explanations. We also hope that the manuscript now fulfills your criteria, and the Journal criteria, for publication.

Reviewer 2 Report

In the present review, the authors aimed to summarize the current findings of circRNA-miRNA-lncRNA networks in the context of sepsis as biomarker and therapeutic target with particular regard to their clinical utility in hospital-setting and their performance in generating clinically actionable data.

The study covers some issues that have been overlooked in other similar topics. The structure of the manuscript appears adequate and well divided in the sections. Moreover, the study is easy to follow, but few issues should be improved. Some of the comments that would improve the overall quality of the study are:

  1. Authors must pay attention to the technical terms acronyms they used in the text.
  2. English language needs to be revised.
  3. Conclusion Section: This paragraph required a general revision to eliminate redundant sentences and to add some "take-home message".

Author Response

REVIEWER #2:

We would like to thank you for your expert review of our manuscript. Thank you very much for your positive opinion regarding our manuscript. We appreciate your opinion very much and believe that your suggestions can help to significantly improve the quality of our review.

In the present review, the authors aimed to summarize the current findings of circRNA-miRNA-lncRNA networks in the context of sepsis as biomarker and therapeutic target with particular regard to their clinical utility in hospital-setting and their performance in generating clinically actionable data.

The study covers some issues that have been overlooked in other similar topics. The structure of the manuscript appears adequate and well divided in the sections. Moreover, the study is easy to follow, but few issues should be improved. Some of the comments that would improve the overall quality of the study are:

Authors must pay attention to the technical terms acronyms they used in the text.

Thank you for the suggestion. All acronyms were evaluated and explained in the text when first used

English language needs to be revised.

We are aware of this limitation, we have provided a complete re-edition of the English language in the revised version of our manuscript

Conclusion Section: This paragraph required a general revision to eliminate redundant sentences and to add some "take-home message".

As recommended, the conclusions have been revised and the most relevant take-home messages have been added

We hope that we have successfully changed our manuscript according to your suggestions and that we have provided all the necessary explanations. We also hope that the manuscript now fulfills your criteria, and the Journal criteria, for publication.

Reviewer 3 Report

The article includes a comprehensive and technically competent overview on the use of non-coding RNA networks as potential novel biomarker and therapeutic target for sepsis and sepsis-related multi-organ failure. While the review is thoroughly and well written, it can be further improved by enhancing its readability.

  1. Please provide a graphical figure which summarizes the cellular mechanisms for the presented novel biomarker.
  2. Summarise the information contained in Table 1.
  3. Please fill in the blank lines in Table 1 (e.g. not applicable).

Author Response

REVIEWER #3:

We would like to thank you for your careful review of our manuscript. Thank you very much for your criticisms and suggestions regarding our manuscript.

The article includes a comprehensive and technically competent overview on the use of non-coding RNA networks as potential novel biomarker and therapeutic target for sepsis and sepsis-related multi-organ failure. While the review is thoroughly and well written, it can be further improved by enhancing its readability.

  1. Please provide a graphical figure which summarizes the cellular mechanisms for the presented novel biomarker.

Thank you for the suggestion. We are aware of the need to make the information provided in our manuscript more easily accessible. Therefore, we have added a figure schematizing the mechanisms by which certain miRNAs may be considered appropriate candidates as novel biomarkers in the septic patient

  1. Summarize the information contained in Table 1.

Table 1 (which in the revised version becomes table 2 ) has been revised, lightened by much information and simplified

  1. Please fill in the blank lines in Table 1 (e.g. not applicable).

Thank you for the suggestion. There are no more blanks in the tables

We hope that we have successfully changed our manuscript according to your suggestions and that we have provided all the necessary explanations. We also hope that the manuscript now fulfills your criteria, and the Journal criteria, for publication.

Round 2

Reviewer 1 Report

Thank you for your effort in reviewing the manuscript. In its current format, it is much friendlier to read and its overall quality has improved.

Author Response

Thank you for your effort in reviewing the manuscript. In its current format, it is much friendlier to read and its overall quality has improved.

We would like to thank again the reviewer whose suggestions made it possible to significantly improve the quality of our manuscript